# Acceleration of western Arctic sea ice loss linked to the Pacific North American pattern

Zhongfang Liu [1 ✉], Camille Risi[2], Francis Codron [3], Xiaogang He[4], Christopher J. Poulsen [5], Zhongwang Wei [6,7], Dong Chen [8], Sha Li[9] & Gabriel J. Bowen [10]

Recent rapid Arctic sea-ice reduction has been well documented in observations, reconstructions and model simulations. However, the rate of sea ice loss is highly variable in both time and space. The western Arctic has seen the fastest sea-ice decline, with substantial interannual and decadal variability, but the underlying mechanism remains unclear. Here we demonstrate, through both observations and model simulations, that the Pacific North American (PNA) pattern is an important driver of western Arctic sea-ice variability, accounting for more than 25% of the interannual variance. Our results suggest that the recent persistent positive PNA pattern has led to increased heat and moisture fluxes from local processes and from advection of North Pacific airmasses into the western Arctic. These changes have increased lower-tropospheric temperature, humidity and downwelling long-wave radiation in the western Arctic, accelerating sea-ice decline. Our results indicate that the PNA pattern is important for projections of Arctic climate changes, and that greenhouse warming and the resultant persistent positive PNA trend is likely to increase Arctic sea-ice loss.

[1] State Key Laboratory of Marine Geology, Tongji University, Shanghai, China. [2] Laboratoire de Météorologie Dynamique, IPSL, CNRS, Sorbonne Université, Paris, France. [3] Laboratoire d'Océanographie et du Climat (LOCEAN), IPSL, CNRS, IRD, Sorbonne Université, Paris, France. [4] Department of Civil and Environmental Engineering, National University of Singapore, Singapore, Singapore. [5] Department of Earth and Environmental Sciences, University of Michigan, Ann Arbor, MI, USA. [6] Guangdong Province Key Laboratory for Climate Change and Natural Disaster Studies, School of Atmospheric Sciences, Sun Yat-sen University, Guangzhou, China. [7] Southern Marine Science and Engineering Guangdong Laboratory (Zhuhai), Zhuhai, China. [8] Nansen-Zhu International Research Centre, Institute of Atmospheric Physics, Chinese Academy of Sciences, Beijing, China. [9] Department of Earth System Science, Tsinghua University, Beijing, China. [10] Department of Geology and Geophysics, University of Utah, Salt Lake City, USA. ✉email: liuzf406@gmail.com

The recent rapid decline in Arctic sea ice has been well documented in observations[1,2], reconstructions[3], and model simulations[4,5]. This decline is strongest in late summer (August–September–October, ASO) with large inter-annual variability[6,7]. The long-term decline in Arctic sea ice has been attributed, in large part, to increasing anthropogenic greenhouse warming and is expected to result in an ice-free Arctic summer within a few decades[4,8]. Arctic sea ice loss is not spatially uniform, with the strongest decline and interannual-to-decadal variability occurring in the western Arctic[9,10] (Fig. 1a and Supplementary Fig. 1). The rapid sea ice decline has led to increased freshwater storage[11], surface water $CO_2$ increase[12] and acidification[13], and phytoplankton blooms[14] in the western Arctic Ocean, as well as frequent extreme weather[15,16] and reduced primary productivity[17] across North America.

Despite their widespread impacts, the mechanisms that drive western Arctic sea ice decline and interannual-to-decadal variability are not well established. Many recent studies have suggested an important role for internal variability of both the atmosphere and ocean[10,18–21], which drives western Arctic sea ice variability dynamically through surface winds and thermodynamically through heat and moisture transport. Recent changes in atmospheric circulation patterns, including the Arctic Oscillation (AO)/North Atlantic Oscillation (NAO)[22], Arctic Dipole (AD)[23], Arctic cyclones[7], and anticyclones[10,24] have been linked to sea ice decline in the region. In particular, an enhanced anticyclonic circulation over Greenland (GL-Z200) has been found to be the main driver of interannual variability and long-term trends in western Arctic sea ice since 1979[10,20]. This anticyclonic circulation is largely generated by tropical Pacific sea surface temperature (SST) forcing through atmospheric teleconnections[25–27]. The link between the tropical Pacific and the Arctic has been referred to as the Pacific–Arctic teleconnection and features enhanced warming and sea ice loss in response to La Niña-like Pacific SST anomaly[27–29] (although this relationship is nonstationary in time[30]). Some other modes of decadal climate variability, such as the Interdecadal Pacific Oscillation (IPO), the Pacific Decadal Oscillation (PDO), and the Atlantic Multidecadal Variability (AMV) have also been shown to drive low-frequency Arctic sea ice fluctuations[21,27,31–35].

In contrast to Arctic atmospheric circulation patterns and Pacific SST forcing, the influence of extratropical teleconnection patterns on western Arctic sea ice has received less attention. Ref. [36] found that the record-breaking 2007 summer western Arctic sea ice decline was accompanied by a strong positive Pacific North American[37] (PNA) pattern, expressed as an anomalous anticyclone over the western Arctic, and argued for an important role of the PNA pattern in driving sea ice loss in the region. However, such strong western Arctic anticyclones are not unprecedented in the instrumental record. For example, a stronger western Arctic anticyclone occurred during the summer of 1987 but was not associated with strong sea ice loss[2,38]. Although more recent work found a statistical link between western Arctic sea ice and PNA trends[21], the physical mechanisms behind this link are not well established. Furthermore, it remains unknown whether the PNA pattern plays a role in the year-to-year variability in western Arctic ice. As a result, it is still an open question whether and how the PNA pattern is systematically linked with interannual to decadal variability in western Arctic sea ice.

In this study, we examine the long-term variability in summer western Arctic sea ice concentration (SIC) in the context of PNA changes to provide a more comprehensive picture of how the PNA pattern affects western Arctic sea ice on interannual-to-decadal timescales. We combine observational satellite data of Arctic SIC, ERA-Interim atmospheric reanalysis, and numerical simulations (see "Methods" section) to demonstrate the importance of the PNA as a driver of western Arctic sea ice loss on both interannual and decadal timescales, and diagnose the associated physical mechanisms.

## Results

**The observed link between western Arctic sea ice and the PNA pattern.** The PNA pattern is a leading mode of Northern Hemisphere atmospheric and climate interannual variability[37]. Although strongest in winter, the PNA also has an atmospheric presence during summer[36], featuring a wave train shifted pole-ward of its wintertime location and a persistent anticyclonic circulation over the western Arctic[36] (see "Methods" section). A positive summer PNA pattern has been associated with enhanced poleward transport of heat and moisture from the North Pacific into the western Arctic[35,36]. The PNA pattern has had a positive trend throughout the satellite era (i.e., since 1979; Fig. 1c) and the spatial pattern of Arctic SIC change that is correlated with PNA change (Fig. 1b) is very similar to the overall pattern of the SIC trends (Fig. 1a), with the fastest sea-ice decline over the western Arctic (including the Beaufort, Chukchi and East Siberian seas).

We construct an index of western Arctic SIC by averaging summer SIC over a region encompassing the highest historical declines (Fig. 1a). The SIC index for the period 1979–2016 shows a significant secular decrease (18% per decade, $p < 0.01$) super-imposed on strong interannual-to-decadal variability, largely in parallel with the evolution of the observed PNA index (see "Methods" section) (Fig. 1c). Lead-lag correlations indicate that the western Arctic SIC is strongly correlated with the simultaneous (ASO) and 1-month leading (JAS) PNA index, and correlates more weakly with the 1-month lagging (SON) PNA index (Supplementary Fig. 2), suggesting that the PNA pattern acts as a precursor to western Arctic SIC changes. The simultaneous correlation is the strongest, corresponding to about 40% ($p < 0.01$) of shared variance. This SIC-PNA correlation is still robust ($r = -0.55$, $p < 0.01$) when the long-term trends are

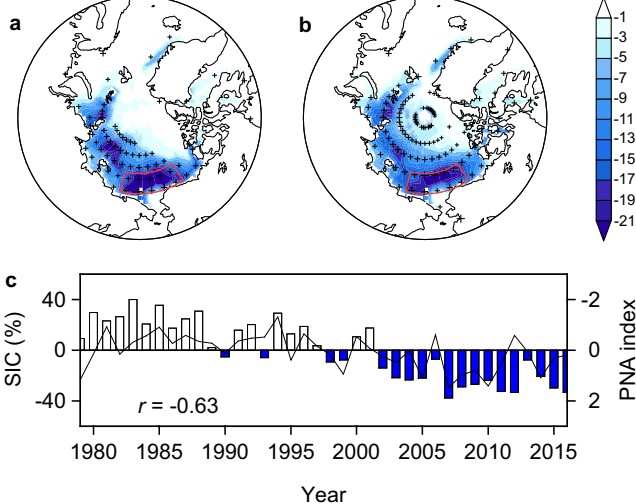

**Fig. 1 Changes in western Arctic sea ice and Pacific North American (PNA) index. a** Linear trend (% per decade) of summer (August–September–October, ASO) sea-ice concentration (SIC) over the period 1979–2016. **b** SIC regressed onto the PNA index (see "Methods" section). **c** Time series of SIC (bar) averaged over the region with the fastest SIC decline (165–215°E and 71–76°N, red box) and PNA index (inverted, curve). The stippling indicates statistical significance at the 5% level in **a** and **b**.

removed (Supplementary Fig. 3). These results suggest that the PNA pattern may be an important driver of western Arctic sea ice variability, both on interannual and decadal timescales.

In contrast, the western Arctic SIC is weakly linked to other large-scale climate modes, especially on interannual timescales (Supplementary Fig. 4). Of the climate indices analyzed, the western Arctic SIC shows significant correlations with the aforementioned three atmospheric modes, namely the GL-Z200 ($r = -0.34$, $p < 0.05$), the AO ($r = 0.29$, $p < 0.1$), and the AD ($r = -0.29$, $p < 0.1$). This is in agreement with the previous studies[10,22,23], but the strength of these correlations is substantially weaker than that with the PNA pattern. The evolution of western Arctic sea ice also shows significant coherence with low-frequency SST variabilities such as the PDO[37] ($r = 0.29$, $p < 0.1$) and the AMV[24] ($r = -0.72$, $p < 0.01$). The AMV, which exhibits the strongest correlation, has previously been shown to drive multidecadal fluctuations in Arctic sea ice[34,39]. The recent trends in western Arctic sea ice may also be associated with the transition of AMV from a negative phase to a positive phase in the mid-1990s. Considering the potential influence of such long-term trends on correlation, we repeat our analysis by linearly detrending the original time series, which results in weak and insignificant correlations between these climate indices and western Arctic SIC (Supplementary Fig. 4). This suggests that these large-scale climate patterns have little influence on the interannual variability of western Arctic sea ice, though they may contribute to its multidecadal declining trend to some extent.

Low SIC is also associated with a coherent anomaly in the atmospheric circulation. Regression of 500-hPa geopotential height (Z500) onto the SIC index reveals a strong anticyclonic anomaly over the western Arctic and a meridional dipole pattern over the North Pacific with a weak cyclonic anomaly over the Aleutians and anticyclonic anomaly over mid-latitudes (Fig. 2a). This circulation pattern is suggestive of a Rossby wave train emanating from the mid-latitude Pacific. The dipole pattern over

the North Pacific extends from the surface to the upper troposphere, depicting a roughly barotropic structure (Supplementary Fig. 5). Such an atmospheric circulation pattern is reminiscent of the summertime PNA pattern described by ref. [36] (see "Methods" section). The pattern of Z500 regressed onto the SIC index bears a striking resemblance to the observed Z500 secular trend over the same period (Fig. 2b), both of which are broadly similar to the Z500 anomaly associated with the positive phase of the PNA pattern (Fig. 2c). This is consistent with the observed correlation between SIC and PNA indices and supports a robust link between western Arctic sea ice decline and a positive PNA-like circulation pattern.

To understand the physical mechanisms linking PNA change and western Arctic sea ice, we examine the interannual variability of PNA-induced thermodynamic processes. During the positive PNA phase, enhanced dipolar pressure anomaly over the North Pacific (Fig. 2a–c) favors poleward transport of heat and moisture from the North Pacific[35,36]. Warm and moist Pacific air is advected into the western Arctic along the west coast of North America, leading to significant increases in lower-tropospheric temperature and humidity in the region (Fig. 2d, e). These warm and humid conditions are further augmented by evaporation from ice-free regions[40,41] and adiabatic subsidence[10,24] due to the anomalous western Arctic anticyclone (Supplementary Fig. 6). Increases in lower-tropospheric temperature and humidity enhance downwelling longwave radiation (DLR; Supplementary Fig. 7a), contributing to western Arctic sea ice decline. These processes are also supported by increased upwelling longwave radiation (ULR) in the western Arctic due to lower sea ice cover and warmer surface temperatures during the positive PNA phase (Supplementary Figs. 7b and 8d). Correlations of the detrended data indicate that the western Arctic temperature, humidity, and DLR are highly correlated with each other ($r = 0.86$ to $0.90$) and they are also significantly correlated with both the PNA index

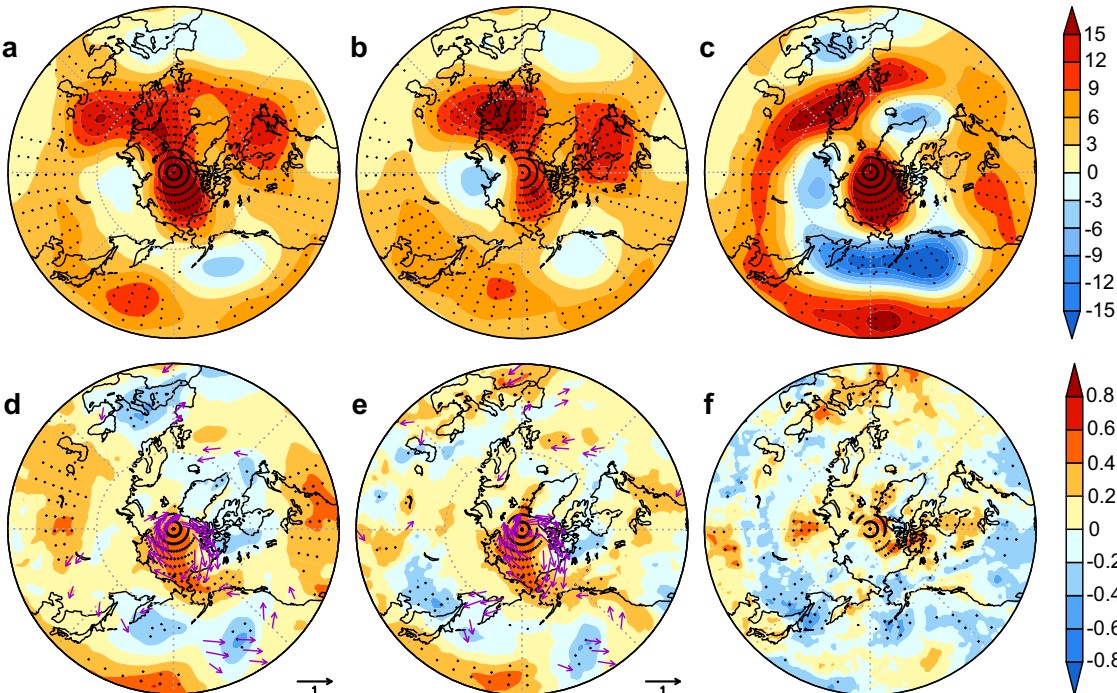

**Fig. 2 PNA-associated atmospheric circulation patterns and thermodynamic impacts. a** 500 hPa geopotential height (Z500, m) regressed onto the standardized region-averaged SIC index. **b** Linear trend (m per decade) of Z500. **c** Z500 (m) regressed onto the PNA index. **d–f** Spatial correlations between the detrended PNA index and thermodynamic variables for lower-tropospheric (1000–850 hPa) temperature (shading) and vertically integrated heat flux (arrows) (**d**), lower-tropospheric (1000–850 hPa) specific humidity (shading), and vertically integrated moisture flux (arrows) **e** and low cloud cover **f**. The stippling indicates statistical significance at the 5% level in all plots.

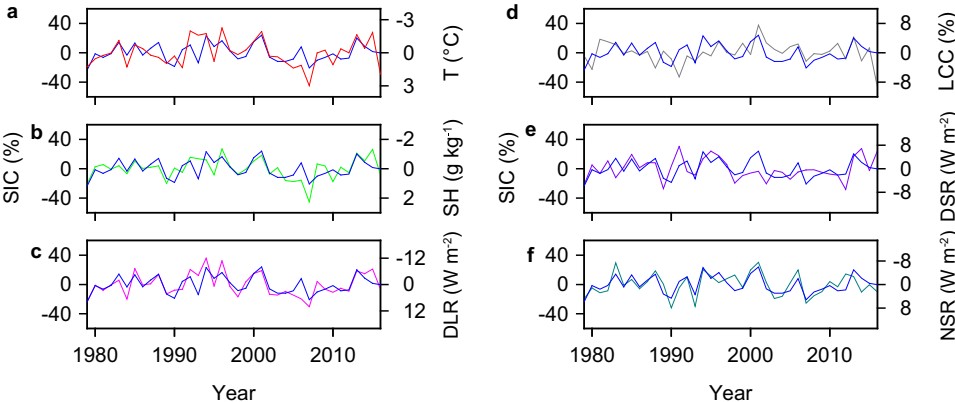

**Fig. 3 PNA-associated thermodynamic impacts on western Arctic SIC. a** Time series of the detrended western Arctic SIC and lower-tropospheric (1000–850 hPa) temperature (T). **b-f** Same as **a** but for lower-tropospheric specific humidity (SH, **b**), downwelling longwave radiation (DLR, **c**), low cloud cover (LCC, **d**), downwelling shortwave radiation (DSR, **e**), and net shortwave radiation (NSR, **f**).

($r = 0.47$ to 0.49; Supplementary Fig. 8) and western Arctic SIC ($r = -0.75$ to $-0.58$; Fig. 3a–c). These intercorrelations further corroborate that the PNA affects western Arctic sea ice variability through its influence on thermodynamic processes.

Previous studies have suggested that the enhanced western Arctic anticyclone during the positive PNA phase reduces cloud cover and thus increases downwelling shortwave radiation (DSR), contributing to sea ice decline[36,38]. Long-term observations indicate that on interannual timescales, low clouds and high clouds over the western Arctic are not sensitive to the PNA, but a positive PNA tends to cause a slight decrease in middle clouds (Fig. 2f, Supplementary Figs. 9a, b, and 10a–c). This decrease in middle clouds, however, is not mirrored in DSR, which shows no correlation with the PNA (Supplementary Figs. 9c and 10d). Observed western Arctic SIC also shows no significant correlation with cloud cover at different levels (Fig. 3d and Supplementary Fig. 11), but exhibits a significant positive correlation with DSR ($r = 0.51$, $p < 0.01$; Fig. 3e), countering expectations that DSR contributes to sea ice decline. We suggest that the counter-intuitive effect of DSR is a result of ice-albedo feedback. During the positive phase of the PNA pattern, anomalously low surface albedo over the western Arctic (Supplementary Figs. 9d and 10e) due to reduced sea ice causes a decrease in the reflection of shortwave radiation from clouds and thus a reduction in DSR[42–44]. This is supported by significant positive net shortwave radiation (NSR) anomalies over the western Arctic during the positive PNA phase (Supplementary Figs. 9e and 10f) and a strong negative correlation between the detrended western Arctic NSR and SIC indices ($r = -0.78$, $p < 0.01$; Fig. 3f). This supports the hypothesis that variation in DSR does not force western Arctic sea ice change but varies as an indirect response to changes in sea ice cover.

Given the strong correlation and mechanistic association between interannual PNA variability and SIC, it seems likely that longer-term changes in the PNA pattern may contribute to trends in western Arctic sea ice. The PNA, both in winter and summer, has exhibited a significant positive trend since the satellite era[45,46], with a magnitude that seems unprecedented over the past millennium[47]. This trend is particularly strong for the summertime PNA during the period 1979–2016 (0.27 per decade, $p < 0.01$), with the pattern shifting from a negative phase in the 1980s through the mid-1990s to a strong positive phase after the mid-1990s (Fig. 1c). These changes have induced significant increases in poleward heat and moisture transport into the western Arctic, leading to enhanced warming and moistening (Supplementary Fig. 12a and b) and thus greater DLR in the

region. Such thermodynamic changes are mirrored by the western Arctic SIC, manifesting as opposite-phase transitions and a rapid decline (Supplementary Fig. 12c). The climatic link between PNA changes and multi-decadal western Arctic sea ice decline is shown by significant correlations between the non-detrended PNA index and lower-tropospheric temperature ($r = 0.60$), humidity ($r = 0.61$), and DLR ($r = 0.61$), and between these parameters and the SIC index ($r = 0.78$, 0.78, and 0.89, respectively).

**Simulated PNA-like atmospheric forcing on western Arctic sea ice.** To complement the observational results and test the plausibility of the proposed SIC-PNA link, we perform two numerical simulations using the Laboratoire de Météorologie Dynamique Zoom (LMDZ) atmospheric general circulation model[48] coupled with a slab ocean sea-ice model[49]. We first run the LMDZ nudged towards observed winds over the period 1979–2016 (see "Methods" section), but with imposed climatological sea surface temperature (SST) and sea ice conditions, as well as fixed $CO_2$ (labeled 'forced-nudged simulation'). Then, in a second simulation, we include these same forcings, but couple the LMDZ to a slab ocean sea-ice model (labeled 'slab-nudged simulation') with a flux correction (see "Methods" section).

Both simulations show Z500 and lower-tropospheric temperature responses to interannual PNA variability that are comparable to the observed patterns (Fig. 4a–d). The model reproduces the amplification of the western Arctic anticyclone and bipolar pressure pattern over the North Pacific, albeit with reduced magnitude (Fig. 4a–c). The associated positive PNA-like pattern results in a significant model warming over the western Arctic, especially in the slab-nudged simulation (Fig. 4c), but the magnitude of these effects is underestimated. The simulated lower-tropospheric temperature time series is highly correlated with observations ($r = 0.86$ and 0.82 for forced-nudged and slab-nudged simulations, respectively; Fig. 4e). This similarity implies that the observed interannual variability in western Arctic lower-tropospheric temperature is largely a consequence of the atmospheric forcing with a strong impact of PNA-like circulation changes.

The impact of atmospheric forcing on interannual variability of western Arctic sea ice is also apparent in the slab-nudged model. The shared variance between simulated and observed SIC indices are 29%, but the model systematically underestimates SIC variability (Fig. 4f). The relevant atmospheric forcing is largely a manifestation of a PNA-like pattern: both the spatial pattern of SIC regressed onto the PNA index and correlation between the

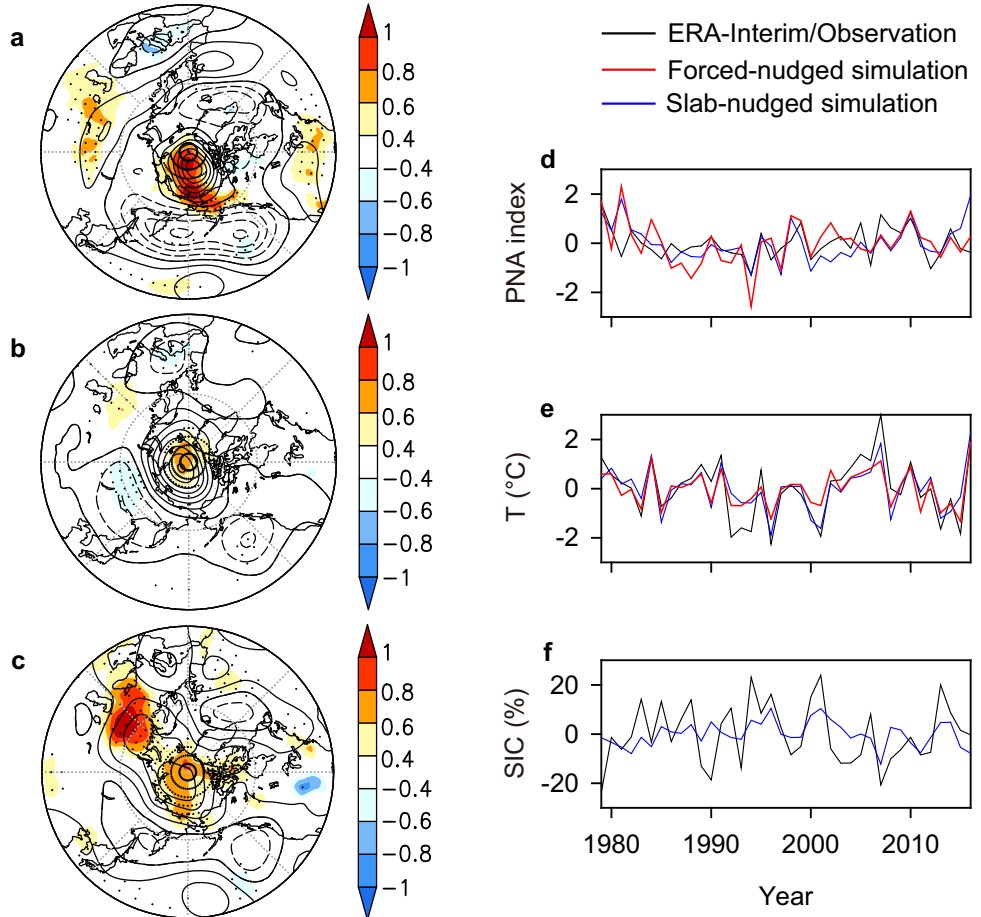

**Fig. 4 Western Arctic SIC responses to interannual PNA variability. a** Regression patterns of the detrended ERA-interim Z500 (contour, m per decade, 25 m interval) and lower-tropospheric (1000–850 hPa) temperature (shading, °C per decade) onto the observed PNA index. **b, c** Same as **a** but for the LMDZ forced-nudged **b** and slab-nudged simulations (**c**). Shading with stippling indicates statistical significance at the 5% level in **a–c. d** Detrended time series of the observed and simulated PNA indices. **e, f** Same as **d** but for lower-tropospheric temperature and SIC.

detrended SIC and PNA indices ($r = -0.51$, $p < 0.01$) (Supplementary Fig. 13) are comparable to the observed counterparts (Supplementary Fig. 3).

On interdecadal timescales, simulated Arctic sea ice response to atmospheric forcing generally mimics the observed spatial pattern and exhibits the strongest decline over the western Arctic, but with a lower magnitude (Fig. 5a vs. Fig. 1a). The derived SIC index indicates that the simulated atmospheric forcing accounts for 56% of the variance in observed western Arctic sea ice (Supplementary Fig. 14), but substantially underestimates the magnitude of western Arctic sea ice decline. This is consistent with the weaker warming over the western Arctic in model simulations than in actual observations (Supplementary Fig. 15). Regression of the simulated Arctic SIC onto the PNA index shows a coherent pattern, with the strongest decline over the western Arctic (Fig. 5a, b). This is generally consistent with observed counterparts (Fig. 1a, b), suggesting that the PNA-like atmospheric forcing is an important driver of the long-term trend and multidecadal variability in western Arctic sea ice. This is further confirmed by the significant negative correlation between the simulated western Arctic SIC and PNA indices ($r = -0.67$, $p < 0.01$; Fig. 5c), which is also comparable to the observed target (Fig. 1c). The associated mechanisms are also clearly visible in the model, in which the positive PNA pattern leads to increases in lower-tropospheric temperature and humidity and thus DLR over

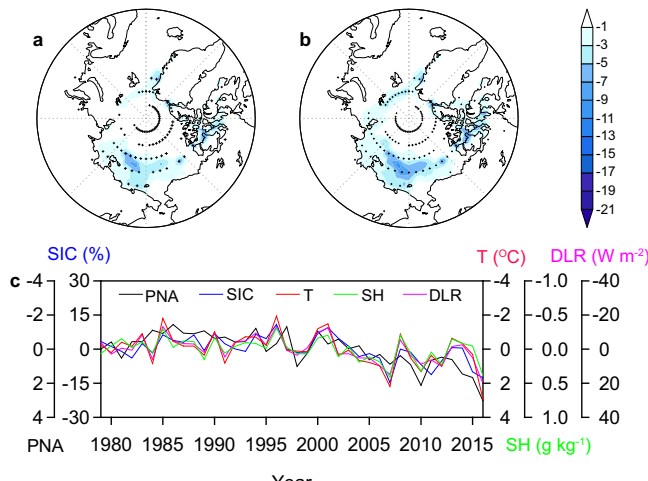

**Fig. 5 Simulated Arctic SIC responses to PNA changes. a** Linear trend (% per decade) of SIC over the period 1979–2016. **b** SIC regressed onto the PNA index. **c** Time series of western Arctic SIC anomaly along with PNA index, lower-tropospheric temperature (T), specific humidity (SH), and downwelling longwave radiation (DLR). Shading with stippling indicates statistical significance at the 5% level in **a** and **b**.

the western Arctic, together with accelerating decline in sea ice in the region (Fig. 5c).

We also explore the robustness of the link between the PNA and western Arctic sea ice in the Coupled Model Intercomparison Project Phase 5 (CMIP5)[50] simulations (see "Methods" section). The CMIP5 multi-model ensemble mean (MMEM) captures the observed declining trend in sea ice across the entire Arctic, but the magnitude is largely underestimated[4], especially in the western Arctic (Supplementary Fig. 16a and c), where the strongest decline in observations is not reproduced (Supplementary Fig. 16a vs. Fig. 1a). Despite these discrepancies, the simulated pattern of sea ice change is similar to the regression pattern of the SIC onto the PNA index (Supplementary Fig. 16a and b), suggesting a broad influence of the positive PNA pattern on a long-term decline in Arctic sea ice. In the western Arctic, this positive PNA trend accounts for 75% of the sea ice decline but contributes little to the interannual variability (Supplementary Fig. 16d). This is not surprising because the PNA is largely a mode of internal climate variability, and the PNA variation in the CMIP5 MMEM dominantly represents a response to changes in external forcing, with short-term internal variability largely canceled out by multi-model averaging.

## Discussion

This work builds on the previous recognition that an extreme positive PNA circulation pattern contributed to the record-breaking 2007 summer Arctic sea ice decline[36] and shows that the PNA pattern is more generally an important driver of western Arctic sea ice variability and trends. Our results from both observations and simulations suggest that recent intensification of the positive PNA pattern has contributed to the accelerated decline of western Arctic sea ice through enhanced poleward heat and moisture transport and thus increased DLR in the region.

Although our study demonstrates that the PNA-like atmospheric forcing is an important driver of western Arctic changes, there are at least three caveats worth noting. First, the substantial underestimation of the magnitude of Arctic sea ice decline in our simulations is not unexpected and highlights the likely contribution of other forcing mechanisms to SIC change. The decline in Arctic sea ice is also driven by the atmosphere–ice-ocean interactions[51,52] and ice drift[53], which are missing in this simple slab ocean model, and anthropogenic warming[54], which was not included in our experiments. Second, the PNA-driven western Arctic ice decline may be also amplified by local feedbacks between the sea ice and atmosphere[52,55]. To isolate the importance of the PNA-like atmospheric forcing from local feedbacks, we compare atmospheric circulation and lower-tropospheric temperature changes between the forced- and slab-nudged simulations. Similar results between the two experiments (Supplementary Fig. 15a and b) suggest that the decline in the western Arctic sea ice is primarily a result of the PNA-like atmospheric forcing rather than a result of local feedbacks of the sea surface conditions on atmospheric temperature or humidity. Our results are broadly consistent with the previous studies[10,51,56] that show a predominant atmospheric forcing of the sea ice variability rather than the converse. Third, the recent decline in western Arctic sea ice is linked to multiple interannual to decadal modes of internal variability[21] (Supplementary Fig. 4). On multidecadal timescales, in particular, western Arctic sea variability is strongly related to the AMV (Supplementary Fig. 4f). By identifying a strong, mechanistic connection between PNA and short-term SIC variability our work indicates the potential for longer-term PNA change to force SIC reduction, but we are not able to exclude the contribution or primacy of other climate patterns in forcing the recent decline in western Arctic sea ice.

The underlying drivers of the unusual positive PNA trend throughout the satellite era (Fig. 1c) remain poorly understood. We speculate that this trend could be a mixed response to natural climate variability[46,57] or enhanced greenhouse warming[47,58], or both. Summertime PNA variability throughout this period is independent of the El Niño/Southern Oscillation (ENSO; Supplementary Fig. 4d). This contradicts the classical view that the PNA originates mainly from ENSO forcing[59], but is supported by results from our models forced with constant SST and previous studies[36,46]. By contrast, the observed PNA index shows a significant relationship with the AMV (Supplementary Fig. 4f), suggesting that the behavior of the AMV may be one important driver of the PNA change[60]. The phase shift to positive AMV in the mid-1990s may have favored the positive phase of the PNA, contributing to the observed PNA trend. This trend may have been further enhanced by anthropogenic climate change, as shown by simulations that project a more positive PNA pattern in response to recent and future greenhouse warming (Supplementary Figs. 16e and 17).

Our work has implications both for the study of past Arctic climate changes and for projections of future Arctic sea ice variability. Proxy reconstructions have revealed substantial interannual to multidecadal variability of the PNA pattern over the past millineum[47,61]. Given the results presented here, such PNA variability is likely to have affected the evolution of Arctic sea ice, implying that the pre-industrial background state of sea ice across this region may have been quite variable, with implications for regional climatic and ecological feedbacks. Recent PNA trends are anomalous in this context, and model projections suggest a robust trend toward a more positive PNA pattern in the twenty-first century in response to anthropogenic greenhouse gas emissions[58] (Supplementary Fig. 17). This positive PNA trend may augment Arctic sea ice decline due to anthropogenic warming, causing more severe ecological and environmental effects.

## Methods

**Data.** The observed monthly SIC is obtained from the National Snow and Ice Data Center (NSIDC) satellite data[62] for the period 1979–2016. We use monthly geopotential height, temperature, specific humidity, cloud, and radiation data from the latest ERA-Interim reanalysis of the European Centre for Medium-Range Weather Forecasts[63] (ECMWF) for the period 1979–2016 at a global 1° × 1° spatial resolution. For comparison, we also analyze the outputs from 12 climate models (using one realization for each model) archived the Coupled Model Intercomparison Project Phase 5 (CMIP5)[50]. We focus our analysis on the period 1979–2016 by combining historical simulations (1979–2005) and Representative Concentration Pathway (RCP) 8.5 simulations (2006–2016). In this study, both observed and simulated anomalies are referenced to climatological means for 1979–2016.

**Atmospheric model.** Model simulations are performed with the LMDZ5A[64] atmospheric general circulation model (AGCM), developed at the Laboratoire de Météorologie Dynamique (LMD). LMDZ5A is the atmospheric component of the Institute Pierre Simon Laplace (IPSL) coupled model IPSL-CM5A[48] that takes part in the Coupled Model Intercomparison Project 5 (CMIP5) program[50]. The model is run with a horizontal resolution of 1.875°in latitude and 3.75°in longitude, with 19 hybrid layers in the vertical.

**Slab ocean sea-ice model.** The slab ocean is used as the surface model coupled to LMDZ5A[64]. To realistically reproduce the Arctic sea-ice variability, a simple thermodynamic sea ice model[49], of comparable complexity to the slab ocean, is used. In the model, over each grid point, the sea ice is represented by a uniform layer of thickness $H_i$ and fractional area $f_i$. It may be covered by a layer of snow. The temperature at the bottom of the ice layer is always equal to the freezing temperature of seawater $T_o$, while the top temperature $T_i$ can vary. The temperature of the snow layer is $T_i$ at the snow-ice interface, and the surface temperature $T_s$ on top (without snow, $T_i = T_s$ the temperature seen by the atmosphere). The temperature profile within the snow or ice layer is assumed to be linear so that the mean ice temperature is $(T_o + T_i)/2$. In the absence of snow, the evolution of $T_i$ is then given by:

$$\rho_i c_i H_i \frac{\partial T_i}{\partial t} = 2(F_{a-i} - F_{i-o}) \tag{1}$$

where $\rho_i$ and $c_i$ are the volumic mass and specific heat capacity of the ice, and $F_{a-i}$ and $F_{i-o}$ are the heat fluxes from the atmosphere to the ice, and from the ice to the ocean. The latter is computed as a conductive flux within the ice layer:

$$F_{i-o} = \frac{\lambda_i}{H_i}(T_i - T_o) \qquad (2)$$

with $\lambda_i$ the thermal conductivity of the ice. In the presence of snow, the surface fluxes go into the snow layer and are replaced at the top of the ice by the conductive flux within the snow.

As soon as the temperature of the slab ocean layer falls below freezing, its temperature is set back to $T_o$, and the resulting energy difference is used to build ice mass and, if ice is already present, to bring the new ice to temperature $T_i$. A set of rules is used to determine how the new mass is divided between extending the fractional area and thickening. If the ocean temperature becomes above freezing, sea ice is melted and the ocean temperature brought back to $T_o$ in a reverse process. If the surface temperature of the snow or ice becomes positive through surface heat fluxes, it is similarly brought back to freezing level, and the energy difference is used to first melt the snow mass, if present, then part of the ice mass. Any energy left after all the ice is melted is used to warm the ocean, ensuring energy conservation. In practice, once the sea ice is present the slab temperature is set to $T_o$ and the different heat fluxes into the ocean (from the atmosphere, ice, or Q-flux) are used to create or melt ice.

**Flux correction.** The slab model needs an additional, seasonally-varying flux correction, usually called "Q-flux", to accurately reproduce the observed pattern of SST and sea ice. The Q-flux represents the convergence of heating by oceanic motion (horizontal and vertical). To estimate the Q-flux, we use the energy budget for the slab of the ocean and the overlying sea ice:

$$\rho c_p H \frac{\partial SST}{\partial t} - \rho_i L \frac{\partial SIV}{\partial t} = F_{surf} + OHT \qquad (3)$$

where $\rho$ and $\rho_i$ are the ocean and ice densities, $L$ the latent heat for freezing, SIV the sea ice volume, $F_{surf}$ is the surface heat fluxes (into the ocean or ice, including the latent heat from snowfall) and OHT is the convergence of energy by the ocean circulation. The Q-flux is then computed by taking the climatology of these different terms (except OHT) from an atmosphere-only simulation using the same model, with prescribed observed SST and sea ice. The seasonal Q-flux is then:

$$Q\text{-flux} = \rho c_p H \frac{\partial \overline{SST}}{\partial t} \rho_i L \frac{\partial \overline{SIV}}{\partial t} - \overline{F}_{surf} \qquad (4)$$

where the overbar denotes a seasonal mean. Note that there a need to know the sea-ice volume, while only sea-ice fraction is usually given. We estimate the daily sea-ice thickness SIH from the daily sea-ice fraction SIF as follows:

$$SIH = \left(2.8 \times SIF^2 + 0.2\right) \times \left(1. + 2.\left(SIF - SIF_{min}\right)\right) \qquad (5)$$

with $SIF_{min}$ the yearly minimum sea-ice fraction at a given location. The yearly maximum sea ice thickness is then 0.6 m where sea ice melts completely in summer and 3 m where the ocean is ice-covered year-round.

**Experimental design.** Two model experiments are conducted to test the observed link between the PNA and western Arctic sea ice and the associated causal relationship. For the first experiment, the LMDZ5A is run forced by a monthly-mean climatology of SST and sea ice based on the Atmospheric Model Intercomparison Project (AMIP) protocol[65] averaged over the 1979–2008 period, as well as constant $CO_2$. To ensure a realistic representation of large-scale atmospheric circulation, horizontal winds fields are nudged by the European Centre for Medium-Range Weather Forecasts (ECMWF) reanalyses[66], in which, the simulated winds are relaxed in each grid point from surface to the top of the atmosphere toward the corresponding reanalysis winds with a relaxation time of 1 h[67]. The simulation is run for 40 years (1977–2016), with the first two years considered as spin-up and thus discarded from the analysis. Here we call this experiment 'forced-nudged simulation'.

The second model experiment is a slab-ocean simulation, in which the above LMDZ5A is coupled to a slab ocean model with a flux correction (referred to as the Q-flux) so that SST and sea ice can respond to the atmospheric forcing through thermodynamic processes. The Q-flux is derived from the forced-nudged simulation described above, with conditions averaged over the 1979–2016 period. The simulation is run from 1979 to 2016, with a 104-year (1875–1978) spin-up simulation that was run until no more trend in SST or sea ice was noticed neither at the global nor at the regional scale. This spin-up simulation was initialized from the forced-nudged simulation and ran with 13 loops of 8 years (e.g., from 1875 to 1882, from 1883 to 1890, and so on, until from 1971 to 1978) nudged by winds from the 1981-1988 period. We call this experiment 'slab-nudged simulation'.

**Summer PNA pattern and PNA index.** The PNA pattern, although most pronounced during boreal winter[37], has an atmospheric presence throughout the entire year with a varying wave train in location and intensity across seasons[36]. The summertime PNA features a wave train that is shifted north of its winter position with a ridge over the western Arctic[36].

The PNA index can be computed using two different methods. The first one is a linear combination of the standardized 500 hPa geopotential height (Z500) anomalies at four centers of action[37]. The second one is based on a rotated empirical orthogonal function[68] (REOF) or EOF[57,69] analysis of Z500 anomaly poleward of 20° latitude for the Northern Hemisphere. It has been suggested that the REOF or EOF method is superior to the pointwise-based analyses because the teleconnection patterns are based on the whole flow field and not just anomalies at a few selected locations[36]. Therefore, we adopt the second method for our analysis. The observed PNA index is obtained from the NCEP Climate Prediction Center (CPC; http://www.cpc.ncep.noaa.gov). For simulated PNA index, we first apply EOF to the summer Z500 anomaly field and then regress summer Z500 onto the principle component (PC) time series to obtain the regression patterns. We compare such patterns with the regression pattern of the observed Z500 against the PNA index time series provided by the CPC. As shown in Supplementary Fig. 18, the first EOF (EOF1) accounts for 23% of the total variance (Supplementary Fig. 18a) and has a regression pattern similar to the observed counterpart (see Supplementary Fig. 18b vs. Fig. 2c), with a pattern correlation of 0.85 over the region 30°–90°N and 90°E–90°W. Therefore, we define the summer PNA pattern as the EOF1. The corresponding PNA index time series are significantly correlated with the CPC PNA index ($r = 0.57$, $p < 0.01$).

## Data availability
ERA-Interim reanalysis data can be obtained from the ECMWF public data sets web interface, http://apps.ecmwf.int/datasets; NSIDC sea ice satellite sea ice can be downloaded from https://nsidc.org/data/nsidc-0116/versions/3. All CMIP5 model output are available from http://esgf-index1.ceda.ac.uk. Other data are available from the corresponding author on request.

## Code availability
The computer codes that support the analysis within this paper are available from the corresponding author on request.

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

## Acknowledgements

This work was supported by the National Natural Science Foundation of China Grants (41876039 and 42025602) to Z.L. Additional support was provided by the US National Science Foundation Grant EAR-1502786 to G.B. The ERA-Interim reanalysis data are obtained from the ECMWF data server and the sea-ice concentration from the NSIDC.

## Author contributions

Z.L. designed the study, conducted the analysis, and led the writing of the manuscript. C.R. and F.C. performed the general circulation model simulations. X.H., Z.W., D.C., and S.L. performed the PNA index calculations and assisted in the interpretation of the data. G.B., C.P., C.R., and X.H. contributed to the discussion of the mechanisms and improvement of the manuscript.

## Competing interests

The authors declare no competing interests.
