## [Peer Review File · Nature Communications]

REVIEWER COMMENTS

Reviewer #1 (Remarks to the Author):

The authors provide the most comprehensive examination to date on the role of the Pacific North American pattern (PNA), and its trends, in interannual-to-decadal sea ice loss. The paper consists of some very convincing and clear analysis that I believe should be published with a few adjustments:

-- It appears the focus is on the PNA trends during the late summer (JAS), so I think lines 167-171 need to be clearer in this regard. I believe the reference (ref 37) refers only to the winter (Dec-Feb) PNA trends. Thus, one question I have and I suspect readers may have is how statistically significant is the PNA trend during the summer?

-- It would be helpful to provide some background and comparison with other major circulation patterns that have been linked to sea ice decline. For example Ding et al. (2017, Nature Climate Change) point to a major anticyclonic circulation near Greenland that contributed to sea ice decline in this region. It would be nice to understand better how to interpret these results in the context of that study ... does it involve a different timescale? Are different regions of sea ice impacted? For future work, could looking at several major models (e.g. NAO) along with the PNA help describe more variation in sea ice and circulation over the Arctic?

-- Where does the 17% number come from in Line 215? I can see that the 56% comes from Supp. Fig. S10, but the basis of this number is not clear to me.

Minor comments:

-- Figure 1. You might want to explicitly note somewhere (in the text or caption) that the PNA index has been inverted, so that negative values lie on the positive side of the axis.

-- line 114: Would change to "Low SIC is also associated with a coherent anomaly in the atmospheric circulation."

-- line 367. You apply both EOF and REOF? This is not clear to me... which one did you end up using?

-- line 373. Is this a spatial correlation or a temporal one?

- Michelle L'Heureux

Reviewer #2 (Remarks to the Author):

The manuscript investigates the effect of the PNA on Arctic summer sea ice variability over the historical period. It uses observational timeseries and two interesting experiments for this analysis. Mainly the focus is on the period 1979-2016. The analysis itself seems interesting and worth publishing. However, the manuscript is lacking in some additional components, which makes it little bit hard to follow and evaluate. In my view, it requires minor revisions before it can be accepted by the journal.

1. My main concern is that the article lacks sufficient and relevant citations to put the work into context of previous studies on the same topic. The introduction misses pretty much all work on Arctic sea ice over the last decade, including several recent (last 2-3 years) studies on the influence of the Pacific on Arctic summer sea ice, and instead focuses heavily on 2000s and early 2010s articles. Recent articles are only limited cited in this manuscript, but still missing the relevant studies on the Pacific influence on Arctic summer sea ice. It's really not my job to point these out, but for starters, Screen and Deser (2019), Ding et al. (2014, 2017, 2019), Baxter et al (2019) and many more should be read, referenced, and contrasted with the study at hand to point

out what is new and what is different here. Many other relevant studies of Arctic sea ice variability over the last decade are also missing and need to be added in the introduction.

Baxter et al., JC, "How tropical Pacific surface cooling contributed to accelerated sea ice melt from 2007 to 2012 as ice is thinned by anthropogenic forcing")

Topal et al. 2020, JC, "An Internal Atmospheric Process Determining Summertime Arctic Sea Ice Melting in the Next Three Decades: Lessons Learned from Five Large Ensembles and Multiple CMIP5 Climate Simulations")

Bonan et al 2020 GRL, "Nonstationary Teleconnection Between the Pacific Ocean and Arctic Sea Ice".

Meehl, G. A., C. T.Chung, J. M.Arblaster, M. M.Holland, and C. M.Bitiz, 2018: Tropical decadal variability and the rate of Arctic sea ice decrease.

Michelle R. McCrystall, J. Scott Hosking, Ian P. White, Amanda C. Maycock, The Impact of Changes in Tropical Sea Surface Temperatures over 1979–2012 on Northern Hemisphere High-Latitude Climate, Journal of Climate.

Shinji Matsumura, Yu Kosaka, Arctic–Eurasian climate linkage induced by tropical ocean variability, Nature Communications.

2. The authors suggest that the circulation pattern that is critical to drive sea ice is the PNA mode. As we know, the PNA favors a same phase oscillation between the tropical SST and high latitude circulation. Considering the occurrence of an increasing trend toward high pressure in the Arctic in the past decade, we should expect to see a tropical SST warming over the Eastern Pacific if the PNA dominates the arctic-tropical teleconnection on low-frequency time scales. However, in observations, we clearly observe a cooling SST trend or no-change over the tropical Eastern Pacific since 2000s. In some of previous studies, this SST cooling is suggested to be a key driver of the high pressure over the Arctic and the teleconnection mode linking these two systems is attributed to a different mode than the PNA (defined as "PARC" in Baxter et al). To reconcile the finding of this study with previous ones, more analyses should be devoted to understand how the PNA index used here is connected to the tropical SST on interannual and interdecadal time scales so that it is more clear to see whether the PNA is indeed the key circulation pattern determining sea ice melting in the Arctic.

3. the modeling approach used here are pretty similar to that in Ding 2017 and the conclusions of the two papers are very consistent. They both use a nudging method +slab ocean/sea ice and point out the importance of DLR in driving sea ice. So I am wondering what's the new value that the current study could additionally add on our understanding of the topic.

4. In Ding et al, 2017, clouds show a different response vertically to high pressure above. Here, TCC is used (I think this is a total cloud index) and it may wash out some significant signals in different levels. So my suggestion is to use clouds over different levels to recalculate their connections with the index.

5. the authors only examine DLR. However, upwelling LW will be increased due to less sea ice coverage and warmer surface. Thus the change of net LW will be largely muted. The authors should not only check DLR but also Upwelling LR.

Reviewer #3 (Remarks to the Author):

Review of "Acceleration of western Arctic sea ice loss linked to the Pacific North American pattern" by Liu et al.

The authors show that the PNA pattern is a driver of western Arctic sea ice variability and that the observed shift to a long-lasting positive phase of the PNA pattern contributes to western Arctic sea ice loss. The authors use both observations and a global climate model to explain the underlying mechanism of increased heat and moisture fluxes primarily from advection of North Pacific air.

The study is presented clearly and concisely, the mechanisms are convincing, the methods are

clearly described and the topic will likely contribute to recent discussions in the field. However, the study lacks relevance caused by a quite narrow perspective and only weak evidence from the modeling part. In the current version, I don't consider the study of extreme importance to researchers in the field.

For this and the reasons detailed below, I cannot recommend publication in Nature Communication at this point, but I encourage the authors to resubmit a revised version if in their interest.

Major comments:

1) ll. 36-38, and ll. 140+, and ll. 219+: The authors state that the PNA pattern is an important driver of western Arctic sea-ice variability, accounting for 26–30% of the interannual variance. I don't see where the numbers 26-30% come from. They only appear in the abstract, not in the main text.

More generally, I question the importance of the found link and the contextualization to other known patterns (e.g. AMV) that drive Arctic sea ice variability. The authors state that the "PNA pattern explains 22 to 24% of the interannual variance in lower-tropospheric temperature, humidity and DLR over the western Arctic, which themselves explain 34%, 37% and 57% of the interannual variance in western Arctic sea ice, respectively". This reads to me that the PNA pattern explains only about 8% of variance in western Arctic sea ice (34% of 23% in case of temperature), which is not much. Although I see that the PNA index shows a similar trend and similar fluctuations as the western Arctic SIC, I am not convinced about the quantitative strength of this relationship and the overall importance for Arctic sea ice variability as a whole. Also, high correlations between two variables are not surprising when their similar trends are not removed (see e.g., l. 108). The study generally convinces me that the PNA pattern is one driver of western Arctic sea ice variability (which I don't find new!), but I still don't know whether it is the main driver. What other drivers exist and how important are they? What explains the other three quarters of western Arctic sea ice variability? In my opinion, the study is too restricted to the one chosen pattern. It could be that other patterns or processes are of similar relevance, or being even more relevant for western Arctic sea ice variability.

In this context, please compare and refer to the following recent studies:

- Zhang et al.: "Variability of Arctic Sea Ice Based on Quantile Regression and the Teleconnection with Large-Scale Climate Patterns", *J. Climate* (2020) 33 (10): 4009–4025.

<https://doi.org/10.1175/JCLI-D-19-0375.1>

- Castruccio et al.: "Modulation of Arctic Sea Ice Loss by Atmospheric Teleconnections from Atlantic Multidecadal Variability", *J. Climate* (2019) 32 (5): 1419–1441.

<https://doi.org/10.1175/JCLI-D-18-0307.1>

- Olonscheck et al.: "Arctic sea-ice variability is primarily driven by atmospheric temperature fluctuations", *Nature Geoscience* (2019), 12: 430-434. <https://doi.org/10.1038/s41561-019-0363-1>

2) ll. 43-45, and ll. 88+: The authors show the climatic link between PNA changes and multi-decadal western Arctic sea ice decline, but the interesting underlying question why the PNA pattern is in an unprecedented positive phase is not investigated or discussed sufficiently. This is only touched very broadly at the end of the study in ll. 259-261. I think that at least a more elaborated discussion on this fundamental aspect is required.

I very much appreciate Figure S12 that shows the projected Z500 changes. However, while the conclusion is prominently used in the abstract, I find the underlying analysis quite weak.

3) I find the evidence from the global climate model not particularly convincing. How good is the IPSL-CM5A model with respect to western Arctic sea ice? The authors show in Figure S10 that the observed and modeled western Arctic SIC is quite different. Also the warming trend in observations is much stronger than in the model (Figure S11), which might explain the weaker model responses. I am missing a careful model evaluation here and would further recommend not to limit the modeling part of this study to a single model. Although I appreciate very much that the authors complement the study with a modeling part, I don't find this part particularly strong and

convincing. I recommend to use many CMIP models or an additional model with a more similar sea ice and warming trend like the observed trends for a more realistic and elaborated analysis of the involved mechanisms.

Minor comments:

1) I. 52: Please provide more updated references here, e.g. Notz et al.: "Arctic Sea Ice in CMIP6", *Geophysical Research Letters*, 2020, <https://doi.org/10.1029/2019GL086749>

2) II. 58-61: This recent reference might be added here: Ouyang et al.: Sea-ice loss amplifies summertime decadal CO₂ increase in the western Arctic Ocean, *Nature Climate Change*, 10, 678-684 (2020), <https://doi.org/10.1038/s41558-020-0784-2>

3) II. 155+: I do not understand the logical links between the variables here. I do not see how a low surface albedo causes a decrease in the reflection of shortwave radiation from clouds. My understanding is that when there is less sea ice, there is more evaporation, more clouds can form and hence more reflection of downwelling SW radiation by clouds might occur. Please explain this mechanism more carefully.

3) I. 183: It is not clear from the main text what LMDZ is. Please explain at first appearance.

4) Figure S1: The color bar is misleading since SIC is a continuous quantity. Your color bar suggests a difference scaling. Please change.

5) Figure S3: The red box in panel a is not mentioned in the caption.

6) Figure S6: Why is there no surface albedo over midlatitude oceans? It shows white, but the color bar does not contain white. Further, the patterns in b (DSR) and d (NSR) are almost identical. This might be linked to some missing surface albedo values. Please check and change.

Revisions to manuscript “Acceleration of western Arctic sea ice loss linked to the Pacific North American pattern (NCOMMS-20-10870-T)”

Dear Editor and Reviewers,

We are grateful for the thorough evaluations of our work provided by the editor and three reviewers, and are happy to see that all reviewers recognized the potential significance and novel contributions of our work. The comments are very constructive and have helped us to increase the clarity and transparency of the revised manuscript. We have thoroughly and carefully addressed each of the comments and provided point-by-point responses. Please note that our replies are marked by blue color and all line numbers in our response refer to the revised manuscript unless otherwise stated.

Reviewer #1 (Remarks to the Author):

The authors provide the most comprehensive examination to date on the role of the Pacific North American pattern (PNA), and its trends, in interannual-to-decadal sea ice loss. The paper consists of some very convincing and clear analysis that I believe should be published with a few adjustments:

We would like to thank the reviewer for devoting the time to review our manuscript and providing the encouraging remarks with very constructive suggestions. These suggestions have been fully incorporated in the revised manuscript. Our responses to the specific comments are as follows.

-- It appears the focus is on the PNA trends during the late summer (JAS), so I think lines 167-171 need to be clearer in this regard. I believe the reference (ref 37) refers only to the winter (Dec-Feb) PNA trends. Thus, one question I have and I suspect readers may have is how statistically significant is the PNA trend during the summer?

This is a good point ignored in the original submission. In the revised manuscript, we have appropriately rephrased this sentence to clarify this point. Related references have also been added to support our statement (see Lines 208-213).

-- It would be helpful to provide some background and comparison with other major circulation patterns that have been linked to sea ice decline. For example, Ding et al. (2017, Nature Climate Change) point to a major anticyclonic circulation near Greenland that contributed to sea ice decline in this region. It would be nice to understand better how to interpret these results in the context of that study ... does it involve a different timescale? Are different regions of sea ice impacted? For future work, could looking at several major models (e.g. NAO) along with the PNA help describe more variation in sea ice and circulation over the Arctic?

We really appreciate the suggestion of the reviewer that adds to the weight of the manuscript. In the revised version, we have attempted to improve the review of the extensive literature on Arctic sea ice variability in the context of large-scale ocean and atmospheric circulation patterns (see Lines 64-81). We add a paragraph to compare the influence of the PNA and other large-scale climate modes on western Arctic sea ice variability (see Lines 132-149; Supplementary Fig. S4). We demonstrate that the PNA is more important to interannual-to-decadal sea ice variability than other major climate patterns in the western Arctic Ocean.

-- Where does the 17% number come from in Line 215? I can see that the 56% comes from Supp. Fig. S10, but the basis of this number is not clear to me.

Apologies for this confusion. Here the “17%” is the slope of linear relationship between the observed and simulated western Arctic sea ice concentration (SIC). It is used here to show that our model substantially underestimates the magnitude of western Arctic sea ice decline. To avoid the confusion, we have removed the number and rephrased this sentence in the revised manuscript (see Lines 255-258).

Minor comments:

-- Figure 1. You might want to explicitly note somewhere (in the text or caption) that the PNA index has been inverted, so that negative values lie on the positive side of the axis.

Thanks for the suggestion. This has been done (see the caption of Figure 1).

-- line 114: Would change to “Low SIC is also associated with a coherent anomaly in the atmospheric circulation.”

Thanks. Changed as suggested (see Line 150).

-- line 367. You apply both EOF and REOF? This is not clear to me.... which one did you end up using?

We apologize for this confusion. In our original manuscript, we use both methods. We perform both EOF and REOF analyses on the summer Z500 anomaly field and then regress the Z500 field onto the PC time series to obtain the regression patterns. We compare such patterns with the regression pattern of the observed Z500 against the PNA index provided by the Climate Prediction Center (CPC). The results indicate that the spatial pattern of the first EOF (EOF1) largely resembles the CPC regression pattern (see Supplementary Fig. S18b vs. Fig. 2c), with a pattern correlation of 0.85 over the region 30°–90°N and 90°E–90°W. Therefore, we define the summer PNA pattern as the EOF1. The corresponding PNA index time series is significantly correlated with the CPC PNA index ($r = 0.57, p < 0.01$). This has been clarified in the

revised version of the manuscript (see Lines 454-463).

-- line 373. Is this a spatial correlation or a temporal one?

This is a temporal correlation, which has been explicitly clarified it in the revised manuscript (see Lines 457-463).

.

Reviewer #2 (Remarks to the Author):

The manuscript investigates the effect of the PNA on Arctic summer sea ice variability over the historical period. It uses observational timeseries and two interesting experiments for this analysis. Mainly the focus is on the period 1979-2016. The analysis itself seems interesting and worth publishing. However, the manuscript is lacking in some additional components, which makes it little bit hard to follow and evaluate. In my view, it requires minor revisions before it can be accepted by the journal.

Firstly, we thank the reviewer for the encouragement and overall positive and constructive reviews. The comments raised have greatly improved the quality of the manuscript and they have been fully incorporated in the revised manuscript. We have addressed these concerns below.

1. My main concern is that the article lacks sufficient and relevant citations to put the work into context of previous studies on the same topic. The introduction misses pretty much all work on Arctic sea ice over the last decade, including several recent (last 2-3 years) studies on the influence of the Pacific on Arctic summer sea ice, and instead focuses heavily on 2000s and early 2010s articles. Recent articles are only limited cited in this manuscript, but still missing the relevant studies on the Pacific influence on Arctic summer sea ice. It's really not my job to point these out, but for starters, Screen and Deser (2019), Ding et al. (2014, 2017, 2019), Baxter et al (2019) and many more should be read, referenced, and contrasted with the study at hand to point out what is new and what is different here. Many other relevant studies of Arctic sea ice variability over the last decade are also missing and need to be added in the introduction.

Baxter et al., JC, “How tropical Pacific surface cooling contributed to accelerated sea ice melt from 2007 to 2012 as ice is thinned by anthropogenic forcing”)

Topal et al. 2020, JC, “An Internal Atmospheric Process Determining Summertime Arctic Sea Ice Melting in the Next Three Decades: Lessons Learned from Five Large Ensembles and Multiple CMIP5 Climate Simulations”)

Bonan et al 2020 GRL, “Nonstationary Teleconnection Between the Pacific Ocean and Arctic Sea Ice”.

Meehl, G. A., C. T.Chung, J. M.Arblaster, M. M.Holland, and C. M.Bitze, 2018: Tropical decadal variability and the rate of Arctic sea ice decrease.

Michelle R. McCrystall, J. Scott Hosking, Ian P. White, Amanda C. Maycock, The Impact of Changes in Tropical Sea Surface Temperatures over 1979–2012 on Northern Hemisphere High-Latitude Climate, Journal of Climate.

Shinji Matsumura, Yu Kosaka, Arctic–Eurasian climate linkage induced by tropical ocean variability, Nature Communications.

We would like to thank the reviewer for raising these issues and the subsequent suggestions. In the revised manuscript, we have improved the review of the extensive

literature on Arctic sea ice variability in the context of large-scale ocean and atmospheric circulation patterns. In particular, we have made our review of the literature more focused on the relevant studies on the influence of Arctic atmospheric circulations and the Pacific SST on Arctic sea ice (see Lines 64-81). By reviewing these relevant studies, we have explicitly addressed the difference and strength of our work relative to these studies (see Lines 82-96).

2. The authors suggest that the circulation pattern that is critical to drive sea ice is the PNA mode. As we know, the PNA favors a same phase oscillation between the tropical SST and high latitude circulation. Considering the occurrence of an increasing trend toward high pressure in the Arctic in the past decade, we should expect to see a tropical SST warming over the Eastern Pacific if the PNA dominates the arctic-tropical teleconnection on low-frequency time scales. However, in observations, we clearly observe a cooling SST trend or no-change over the tropical Eastern Pacific since 2000s. In some of previous studies, this SST cooling is suggested to be a key driver of the high pressure over the Arctic and the teleconnection mode linking these two systems is attributed to a different mode than the PNA (defined as “PARC” in Baxter et al). To reconcile the finding of this study with previous ones, more analyses should be devoted to understand how the PNA index used here is connected to the tropical SST on interannual and interdecadal time scales so that it is more clear to see whether the PNA is indeed the key circulation pattern determining sea ice melting in the Arctic.

Thank you for bringing this issue to our attention. As the reviewer pointed out, the Pacific-Arctic (PARC) teleconnection¹ is associated with a barotropic anticyclone over northeastern Canada and Greenland, which has been shown to contribute to accelerated summer Arctic sea ice decline in recent decades^{2,3}. The summertime PNA pattern is different from the PARC teleconnection. It features a wave train that is shifted north of its winter position with a persistent anticyclone over the western Arctic⁴. Studies indicate that the PNA components have a seasonal dependence in response to tropical Pacific SST forcing⁵. Although there is a preference for positive wintertime PNA due to El Niño-like warming in the tropical Pacific⁶, Summertime PNA variability throughout our study period is largely independent of the El Niño/Southern Oscillation (ENSO; Supplementary Fig. S4d). This is supported by results from our models forced with constant SST and previous studies^{4,5,7,8}. Therefore, reduced western Arctic sea ice can be driven by both positive summertime PNA and PARC patterns and they do not contradict each other. Here we show observational and modeling evidence that the PNA pattern has more significant influence on western Arctic sea ice than ENSO-like SST forcing.

In the revised version, we have acknowledged the influence of the Pacific-Arctic (PARC) teleconnection on Arctic sea ice (see Lines 73-78) and provided a discussion to reconcile the influence of the PNA and tropical SST forcing on western Arctic sea ice variability (see Lines 314-318).

3. the modeling approach used here are pretty similar to that in Ding 2017 and the conclusions of the two papers are very consistent. They both use a nudging method +slab ocean/sea ice and point out the importance of DLR in driving sea ice. So I am wondering what's the new value that the current study could additionally add on our understanding of the topic.

We acknowledge this point – the reviewer is correct that our experiment design is similar to Ref², and both studies highlight the importance of DLR in driving Arctic sea ice variability. We would assert that the outstanding feature of this work is not revealing new dynamics/thermodynamics of summer Arctic sea decline, which has been widely investigated by many previous studies. Instead, our contribution is to demonstrate that the PNA pattern is an important driver of interannual-to-decadal variability and trends in western Arctic sea ice, and to reveal how changes in the PNA affect western Arctic sea ice through their influence on thermodynamic processes. These are not addressed in Ref² or any other studies. In addition, the study area (western Arctic Ocean), atmospheric circulation pattern (PNA) concerned, as well as the GCM model (LMDZ) used in our work are all different from those in Ref². In particular, we also find that changes in cloudiness are not a contributor to western Arctic sea ice variation, which contrasts Ref². Thus, we believe that these features have additionally added weight of regional Arctic climate research, and could advance our understanding of the Arctic climate changes.

4. In Ding et al, 2017, clouds show a different response vertically to high pressure above. Here, TCC is used (I think this is a total cloud index) and it may wash out some significant signals in different levels. So my suggestion is to use clouds over different levels to recalculate their connections with the index.

Thank you for the suggestion. As the reviewer pointed out, only TCC is used for analysis in our original submission. In the revised version, we have explored the associations of the PNA and western Arctic sea ice with clouds at different levels (low clouds, middle clouds and high clouds). We believe that these new comparisons have increased the clarity and transparency of the revised manuscript while they do not affect our initial results and interpretations (see Lines 187-194 and the related figures).

5. the authors only examine DLR. However, upwelling LW will be increased due to less sea ice coverage and warmer surface. Thus the change of net LW will be largely muted. The authors should not only check DLR but also Upwelling LR.

Thank you for this suggestion which we have followed up on in our revision. We have included additional figures (Supplementary Figs. S7b and S8d) to show the responses of upwelling longwave radiation (ULR) to changes in the PNA pattern and western Arctic sea ice. As the reviewer pointed out, during the positive PNA phase, warmer

surface temperatures and lower sea ice cover over the western Arctic Ocean have contributed to increased ULR (see Lines 176-179).

Reviewer #3 (Remarks to the Author):

Review of “Acceleration of western Arctic sea ice loss linked to the Pacific North American pattern” by Liu et al.

The authors show that the PNA pattern is a driver of western Arctic sea ice variability and that the observed shift to a long-lasting positive phase of the PNA pattern contributes to western Arctic sea ice loss. The authors use both observations and a global climate model to explain the underlying mechanism of increased heat and moisture fluxes primarily from advection of North Pacific air.

The study is presented clearly and concisely, the mechanisms are convincing, the methods are clearly described and the topic will likely contribute to recent discussions in the field. However, the study lacks relevance caused by a quite narrow perspective and only weak evidence from the modeling part. In the current version, I don't consider the study of extreme importance to researchers in the field.

For this and the reasons detailed below, I cannot recommend publication in Nature Communication at this point, but I encourage the authors to resubmit a revised version if in their interest.

We thank the reviewer for the insightful and critical comments. The reviewer's comments have been fully incorporated in the revised manuscript. Among the most significant changes made include the addition of 1) a review of the extensive literature on Arctic sea ice variability in the context of large-scale ocean and atmospheric circulation patterns, 2) an analysis of the relative importance of the PNA and other large-scale climate patterns in driving western Arctic sea ice variability, 3) additional modelling evidence from the CMIP5 multi-model ensemble mean that the PNA pattern affects western Arctic sea ice decline, and 3) a more elaborated discussion of why the PNA pattern has been in an unusual upward trend since 1979.

We believe that these changes have strongly improved the clarity of the manuscript and hope to have addressed all of the concerns raised in the review. Our responses to the specific comments are as follows:

Major comments:

1) ll. 36-38, and ll. 140+, and ll. 219+: The authors state that the PNA pattern is an important driver of western Arctic sea-ice variability, accounting for 26–30% of the interannual variance. I don't see where the numbers 26-30% come from. They only appear in the abstract, not in the main text.

We apologize for the confusion. Here we mean that the PNA pattern accounts for 26% and 30% of the interannual variance in observed and simulated western Arctic sea ice,

respectively. This sentence has been rephrased in the revised manuscript (see Lines 38-39).

More generally, I question the importance of the found link and the contextualization to other known patterns (e.g. AMV) that drive Arctic sea ice variability. The authors state that the “PNA pattern explains 22 to 24% of the interannual variance in lower-tropospheric temperature, humidity and DLR over the western Arctic, which themselves explain 34%, 37% and 57% of the interannual variance in western Arctic sea ice, respectively”. This reads to me that the PNA pattern explains only about 8% of variance in western Arctic sea ice (34% of 23% in case of temperature), which is not much. Although I see that the PNA index shows a similar trend and similar fluctuations as the western Arctic SIC, I am not convinced about the quantitative strength of this relationship and the overall importance for Arctic sea ice variability as a whole. Also, high correlations between two variables are not surprising when their similar trends are not removed (see e.g., l. 108). The study generally convinces me that the PNA pattern is one driver of western Arctic sea ice variability (which I don’t find new!), but I still don’t know whether it is the main driver. What other drivers exist and how important are they? What explains the other three quarters of western Arctic sea ice variability? In my opinion, the study is too restricted to the one chosen pattern. It could be that other patterns or processes are of similar relevance, or being even more relevant for western Arctic sea ice variability.

In this context, please compare and refer to the following recent studies:

- Zhang et al.: “Variability of Arctic Sea Ice Based on Quantile Regression and the Teleconnection with Large-Scale Climate Patterns”, J. Climate (2020) 33 (10): 4009–4025. <https://doi.org/10.1175/JCLI-D-19-0375.1>
- Castruccio et al.: “Modulation of Arctic Sea Ice Loss by Atmospheric Teleconnections from Atlantic Multidecadal Variability”, J. Climate (2019) 32 (5): 1419–1441. <https://doi.org/10.1175/JCLI-D-18-0307.1>
- Olonscheck et al.: “Arctic sea-ice variability is primarily driven by atmospheric temperature fluctuations”, Nature Geoscience (2019), 12: 430-434. <https://doi.org/10.1038/s41561-019-0363-1>

We really appreciate these concerns and suggestions of the reviewer that help increase the clarity and transparency of the revised manuscript, and add to the overall value of the manuscript.

There are two main points here. With respect to the first (“...the study is too restricted to the one chosen pattern...”), we have improved the review of the extensive literature on Arctic sea ice variability in the context of large-scale ocean and atmospheric circulation patterns (see Lines 64-81). Based on the literature review, we have included a comparison of the influence of the PNA to other large-scale climate modes (including AMO) on western Arctic sea ice variability (see Lines 132-149). We demonstrate that the PNA pattern plays a greater role in western Arctic sea ice

variability than other climate modes on interannual timescales, though the AMO has the strongest influence on multi-decadal declining trend in western Arctic sea ice. Lastly, we also add some discussion of potential contribution or primacy of other climate patterns to the recent decline in western Arctic sea ice (see Lines 310-318).

With respect to the second point (“...This reads to me that the PNA pattern explains only about 8% of variance in western Arctic sea ice (34% of 23% in case of temperature), which is not much...”), we apologize for unclear language that may have caused some confusion. Our observations have demonstrated that the PNA explains 30% of the interannual variance in western Arctic sea ice (Lines 127-129 and Supplementary Fig. S3). Here we show the intercorrelations of the PNA index and the western Arctic SIC with lower-tropospheric temperature, humidity and DLR in order to demonstrate that the PNA affects western Arctic sea ice largely through its effects on some thermodynamic processes (including lower tropospheric temperature, humidity, DLR, etc.). It should be noted that one cannot simply multiply these variances to infer the explained variance between the PNA and western Arctic SIC because the relationships of the PNA and western Arctic SIC with these thermodynamic variables are not necessarily independent. For example, in the case of temperature, the PNA explains 22% of the temperature variance, and the temperature explains 34% of the western Arctic SIC variance, but this doesn't necessarily mean that the PNA explains 22% of 34% (around 8%) of the western Arctic SIC variance. Even if we could, it should take at least three thermodynamic variables (temperature, humidity and DLR) into account because the PNA affects western Arctic SIC through these mediating variables. That is to say, the effects need to be added together, or the effects are at least sub-additive, i.e., the maxima of the three effects. In fact, if we calculate the effects of the PNA on the western Arctic SIC using the method suggested by the reviewer, it explains about 29.5% ($0.22*0.34+0.24*0.37+0.23*0.57$) of the interannual variance, which is comparable to the correlation between the PNA index and western Arctic SIC. To avoid confusion, we have rephrased these sentences in the revised manuscript (see Lines 179-184).

2) ll. 43-45, and ll. 88+: The authors show the climatic link between PNA changes and multi-decadal western Arctic sea ice decline, but the interesting underlying question why the PNA pattern is in an unprecedented positive phase is not investigated or discussed sufficiently. This is only touched very broadly at the end of the study in ll. 259-261. I think that at least a more elaborated discussion on this fundamental aspect is required.

This is a good point ignored in the original submission. In the revised manuscript, we have added some discussion on why the PNA pattern has been in an unusual positive trend (see Lines 319-333 for details).

I very much appreciate Figure S12 that shows the projected Z500 changes. However, while the conclusion is prominently used in the abstract, I find the underlying analysis

quite weak.

We thank the reviewer for the encouragement. Please see the response above.

3) I find the evidence from the global climate model not particularly convincing. How good is the IPSL-CM5A model with respect to western Arctic sea ice? The authors show in Figure S10 that the observed and modeled western Arctic SIC is quite different. Also the warming trend in observations is much stronger than in the model (Figure S11), which might explain the weaker model responses. I am missing a careful model evaluation here and would further recommend not to limit the modeling part of this study to a single model. Although I appreciate very much that the authors complement the study with a modeling part, I don't find this part particularly strong and convincing. I recommend to use many CMIP models or an additional model with a more similar sea ice and warming trend like the observed trends for a more realistic and elaborated analysis of the involved mechanisms.

We fully understand the concerns of the reviewer and appreciate the subsequent suggestions. We acknowledge that there do exist some discrepancies between the observed and modeled temperature and SIC – the magnitudes are systematically underestimated. These discrepancies may arise largely due to the model limitations and our experiment design. Our main purpose is to isolate the impacts of atmospheric forcing on Arctic sea ice. To achieve this, we use an atmospheric general circulation model (LMDZ5A, which is the atmospheric component of the IPSL-CM5A) coupled with a slab ocean model. The atmosphere–ice–ocean interactions are missing in this simple slab ocean model, and anthropogenic warming was not included in our experiments, both of which can lead to the underestimation of temperature and SIC. We have discussed caveats associated with the model experimental set-up in our manuscript (see Lines 296-301).

Despite this model-observation discrepancy, the spatial and temporal patterns of atmospheric circulation, temperature ($r = 0.83$ for non-detrended timeseries) and SIC ($r = 0.75$ for non-detrended timeseries) are largely consistent between observations and models (Fig. 4 and Supplementary Figs. S14 and Fig. S15). This suggests that our models are capable of capturing the relevant processes.

We performed additional analysis following follow the reviewer's suggestion. We examined the response of Arctic sea ice to change in PNA using the CMIP5 multimodel ensemble mean (MEM). We find that the MEM captures the observed declining trend in sea ice across the entire Arctic, but the magnitude is largely underestimated⁹, especially in the western Arctic (Supplementary Fig. S16a and c), where the strongest decline in observations is not reproduced (Supplementary Fig. S16a vs. Fig. 1a). Despite these discrepancies, the simulated pattern of sea ice change is similar to the regression pattern of the SIC onto the PNA index (Supplementary Fig. S16a and b), suggesting a broad influence of the positive PNA pattern on long-term

decline in Arctic sea ice. In the western Arctic, this positive PNA trend accounts for 75% of the sea ice decline, but contributes little to the interannual variability (Supplementary Fig. S16d). This is not surprising because the PNA is largely a mode of internal climate variability, and the PNA variation in the CMIP5 MMEM dominantly represents a response to changes in external forcing, with short-term internal variability largely cancelled out by multi-model averaging (see Lines 271-285).

Minor comments:

1) l. 52: Please provide more updated references here, e.g. Notz et al.: “Arctic Sea Ice in CMIP6”, *Geophysical Research Letters*, 2020, <https://doi.org/10.1029/2019GL086749>

Thanks, we have updated the references in the revised manuscript.

2) ll. 58-61: This recent reference might be added here: Ouyang et al.: Sea-ice loss amplifies summertime decadal CO₂ increase in the western Arctic Ocean, *Nature Climate Change*, 10, 678-684 (2020), <https://doi.org/10.1038/s41558-020-0784-2>

Thanks, we have rephrased this sentence and added the suggested references (see Lines 59-62).

3) ll. 155+: I do not understand the logical links between the variables here. I do not see how a low surface albedo causes a decrease in the reflection of shortwave radiation from clouds. My understanding is that when there is less sea ice, there is more evaporation, more clouds can form and hence more reflection of downwelling SW radiation by clouds might occur. Please explain this mechanism more carefully.

Thanks for bringing this point up. We fully understand the concerns and suggested logical links by the reviewer. If these logical links hold true, the cloud cover should increase and the surface net shortwave radiation (NSR) should decrease, but this is not the case in observations. As shown in Figure R1, western Arctic sea ice decreases (Fig. R1b) in response to increased PNA index (Fig. R1a), but this decline in sea ice does not cause an increase in cloud cover (Fig. R1c) due to stronger evaporation. Therefore, reduced downwelling shortwave radiation (DSR; Fig. R1e) may not arise from more reflection by clouds, which is supported by increased NSR (Fig. R1f). Thus, we speculate that other mechanisms may be responsible for the decreased DSR and increased NSR in the western Arctic. These processes may be due to the greatly reduced shortwave radiation reflected back upward at the surface, which can be realized through the surface albedo and cloud feedbacks. The shortwave radiation that clouds reflect includes two parts: one is emitted directly from the sun, and the other is the upward shortwave radiation that is reflected by the surface or the clouds below.

Therefore, the DSR at the surface is the sum of the directly transmitted and reflected radiation. During the positive PNA phase, western Arctic sea ice decrease rapidly, causing a significant decline in surface albedo (AL; Fig. R1d). The lower the surface albedo, the less DSR is reflected back to the atmosphere and the less is re-reflected back to the surface by the clouds. These processes can lead to decreased DSR but increased NSR. This mechanism is also supported by previous studies¹⁰⁻¹².

Figure R1. PNA-induced sea ice variability, cloud and radiation in the western Arctic Ocean. **a**, Time series of the PNA index. **b-f**, Same as **a** but for sea ice (SIC, **b**), cloud cover (TCC, **c**), surface albedo (AL, **d**), downwelling longwave radiation (DLR, **e**) and net shortwave radiation (NSR, **f**).

3) l. 183: It is not clear from the main text what LMDZ is. Please explain at first appearance.

Thanks, we have spelled out the full name of LMDZ at its first appearance in the revised manuscript (see Lines 225-226).

4) Figure S1: The color bar is misleading since SIC is a continuous quantity. Your color bar suggests a difference scaling. Please change.

Thanks, we have made the suggested change in the revised manuscript.

5) Figure S3: The red box in panel a is not mentioned in the caption.

Thanks, this has been clarified in the figure caption in the revised manuscript.

6) Figure S6: Why is there no surface albedo over midlatitude oceans? It shows white, but the color bar does not contain white. Further, the patterns in b (DSR) and d (NSR)

are almost identical. This might be linked to some missing surface albedo values. Please check and change.

Thanks for your concerns about this. We double checked ERA-Interim surface albedo data and our calculations. The albedo is constant over ocean and is approximately 0.06, which causes the correlation between the PNA and surface albedo over midlatitude oceans to be zero (white areas in Supplementary Fig. S9d). This also causes nearly identical pattern between DSR and NSR shown in Supplementary Figure S9c and e. Please note that all figures mentioned here are referred to those in the revised version.

References.

- 1 Baxter, I. *et al.* How tropical Pacific surface cooling contributed to accelerated sea ice melt from 2007 to 2012 as ice is thinned by anthropogenic forcing. *J. Clim.* **32**, 8583-8602 (2019).
- 2 Ding, Q. *et al.* Influence of high-latitude atmospheric circulation changes on summertime Arctic sea ice. *Nat. Clim. Change* **7**, 289 (2017).
- 3 Ding, Q. *et al.* Fingerprints of internal drivers of Arctic sea ice loss in observations and model simulations. *Nat. Geosci.* **12**, 28-33 (2019).
- 4 L'Heureux, M. L., Kumar, A., Bell, G. D., Halpert, M. S. & Higgins, R. W. Role of the Pacific - North American (PNA) pattern in the 2007 Arctic sea ice decline. *Geophys. Res. Lett.* **35** (2008).
- 5 Leathers, D. J. & Palecki, M. A. The Pacific/North American teleconnection pattern and United States climate. Part II: temporal characteristics and index specification. *J. Clim.* **5**, 707-716 (1992).
- 6 Horel, J. D. & Wallace, J. M. Planetary-scale atmospheric phenomena associated with the Southern Oscillation. *Mon. Weather Rev.* **109**, 813-829 (1981).
- 7 Hu, C. *et al.* Shifting El Niño inhibits summer Arctic warming and Arctic sea-ice melting over the Canada Basin. *Nat commun.* **7**, 1-9 (2016).
- 8 Weng, H., Ashok, K., Behera, S. K., Rao, S. A. & Yamagata, T. Impacts of recent El Niño Modoki on dry/wet conditions in the Pacific rim during boreal summer. *Clim. Dyn.* **29**, 113-129 (2007).
- 9 Stroeve, J. C. *et al.* Trends in Arctic sea ice extent from CMIP5, CMIP3 and observations. *Geophys. Res. Lett.* **39** (2012).
- 10 Francis, J. A., Hunter, E., Key, J. R. & Wang, X. Clues to variability in Arctic minimum sea ice extent. *Geophys. Res. Lett.* **32** (2005).
- 11 Kapsch, M.-L., Graverson, R. G., Tjernström, M. & Bintanja, R. The effect of downwelling longwave and shortwave radiation on Arctic summer sea ice. *J. Clim.* **29**, 1143-1159 (2016).
- 12 Krikken, F. & Hazeleger, W. Arctic energy budget in relation to sea ice variability on monthly-to-annual time scales. *J. Clim.* **28**, 6335-6350 (2015).

REVIEWERS' COMMENTS

Reviewer #1 (Remarks to the Author):

I read through the edits and responses and am pleased with the author's changes. I think this manuscript is in good enough condition to be published.

Reviewer #2 (Remarks to the Author):

I think the authors have done a lot of additional work and re-work on this manuscript to address all concerns I brought up in the first round. Thus, my recommendation is that the paper be accepted for publication.

Reviewer #3 (Remarks to the Author):

I congratulate the authors for this very careful revision of the manuscript. They have satisfactorily addressed all my comments and concerns. I can now recommend publication of the article in Nature Communications.

Dirk Olonscheck

Responses to Reviewers

Reviewer #1 (Remarks to the Author):

I read through the edits and responses and am pleased with the author's changes. I think this manuscript is in good enough condition to be published.

We thank the reviewer for their very helpful suggestions throughout the review process and their kind recommendation.

Reviewer #2 (Remarks to the Author):

I think the authors have done a lot of additional work and re-work on this manuscript to address all concerns I brought up in the first round. Thus, my recommendation is that the paper be accepted for publication.

We thank the reviewer for their very helpful suggestions throughout the review process and their kind recommendation.

Reviewer #3 (Remarks to the Author):

I congratulate the authors for this very careful revision of the manuscript. They have satisfactorily addressed all my comments and concerns. I can now recommend publication of the article in Nature Communications.

Dirk Olonscheck

We thank the reviewer for their very helpful suggestions throughout the review process and their kind recommendation.